# Tankyrase inhibition sensitizes cells to CDK4 blockade

Miguel Foronda[1], Yusuke Tarumoto[2], Emma M. Schatoff[1,3], Benjamin I. Leach[1¤], Bianca J. Diaz[1], Jill Zimmerman[1], Sukanya Goswami[1], Michael Shusterman[1], Christopher R. Vakoc[2], Lukas E. Dow[1,4,5]*

**1** Sandra and Edward Meyer Cancer Center, Weill Cornell Medicine, New York, NY, United States of America, **2** Cold Spring Harbor Laboratory, Cold Spring Harbor, NY, United States of America, **3** Tri-Institutional MD-PhD program, Weill Cornell Medicine, New York, NY, United States of America, **4** Department of Medicine, Weill Cornell Medicine, New York, NY, United States of America, **5** Department of Biochemistry, Weill Cornell Medicine, New York, NY, United States of America

¤ Current address: Dept of Medical Oncology and Therapeutics Research, City of Hope, CA, United States of America
* lud2005@med.cornell.edu

**Data Availability Statement:** All relevant data are within the manuscript and its Supporting Information files. The raw data is under submission and will be released publicly during revision of the manuscript.

## Abstract

Tankyrase (TNKS) 1/2 are positive regulators of WNT signaling by controlling the activity of the ß-catenin destruction complex. TNKS inhibitors provide an opportunity to suppress hyperactive WNT signaling in tumors, however, they have shown limited anti-proliferative activity as a monotherapy in human cancer cell lines. Here we perform a kinome-focused CRISPR screen to identify potential effective drug combinations with TNKS inhibition. We show that the loss of CDK4, but not CDK6, synergizes with TNKS1/2 blockade to drive G1 cell cycle arrest and senescence. Through precise modelling of cancer-associated mutations using cytidine base editors, we show that this therapeutic approach is absolutely dependent on suppression of canonical WNT signaling by TNKS inhibitors and is effective in cells from multiple epithelial cancer types. Together, our results suggest that combined WNT and CDK4 inhibition might provide a potential therapeutic strategy for difficult-to-treat epithelial tumors.

## Introduction

Almost all colorectal cancers (CRCs) carry activating mutations in the WNT pathway, most often due to loss-of-function truncations in the adenomatous polyposis coli (APC) tumor suppressor[1, 2]. Numerous studies have shown that hyperactivation of the WNT pathway is an oncogenic driver in the intestine[3–5], and in many cases, suppression of WNT signaling is sufficient to induce cell differentiation[6] and tumor regression in animal models[7–12]. We recently showed that restoring endogenous regulation of the WNT pathway by re-expressing normal levels of APC, is sufficient to induce disease regression in established colorectal cancers[8, 9].

Efforts to target WNT signaling pharmacologically have explored multiple nodes of inhibition, including blocking WNT ligand secretion and direct targeting the downstream

**Funding:** This work was supported by a project grant from the NIH/NCI (CA195787-01) and a Stand Up to Cancer Colorectal Cancer Dream Team Translational Research Grant (SU2C-AACR-DT22-17). Stand Up to Cancer is a program of the Entertainment Industry Foundation. Research grants are administered by the American Association for Cancer Research, a scientific partner of SU2C. LED was supported by a K22 Career Development Award from the NCI/NIH (CA 181280-01). The content is solely the responsibility of the authors and does not necessarily represent the official views of the NIH. The funders had no role in study design, data collection and analysis, decision to publish, or preparation of the manuscript.

**Competing interests:** LED is a scientific advisory board member and stockholder in Mirimus Inc., who have licensed shRNA technology related to this study. This does not alter our adherence to PLOS ONE policies on sharing data and materials. All other authors declare no competing financial interests.

transcriptional effector, ß-catenin[13, 14]. We reasoned that modulating WNT signaling through the same mechanism by which APC acts would provide the best opportunity for well-controlled WNT regulation. While it is not feasible to genetically restore APC in human CRCs, it is possible to reinforce the activity of the APC/AXIN/ß-catenin destruction complex by increasing the levels of the scaffold proteins AXIN1/2. Tankyrase enzymes (TNKS and TNKS2) are functionally-redundant members of the poly-ADP ribose polymerase (PARP) family that mediate the PARsylation and degradation of AXIN1/2, and thus, TNKS inhibition stabilizes AXIN and reinforces endogenous suppression of WNT signaling. Numerous small molecules have been described that enable selective TNKS1/2 blockade and WNT pathway suppression in CRC cell lines [15–18].

Despite their ability to dampen hyperactive WNT and suppress tumor growth in model systems, TNKS inhibitors do not always show anti-proliferative effects as single agents in human CRC cell lines[17, 19, 20]. However, when mitogenic signaling is limited by the reduction of serum in the culture media, TNKS inhibition can block growth and proliferation of otherwise resistant cells[16, 21]. This suggests that inhibition of parallel mitogenic signals may synergize with TNKS inhibitors to control tumor cell growth. Indeed, blocking MEK, PI3K, or EGFR can enhance the anti-tumor effect of TNKS inhibitors[17, 19, 20]. Here we perform a domain-focused kinome CRISPR library screen[22], aiming to identify the most critical mitogenic signaling components that mediate resistance to TNKS inhibition, with the goal of defining combination approaches to enhance tumor cell response. We identify CDK4 as a synthetic vulnerability in colorectal cancer cells and show that dual inhibition of TNKS and CDK4 drives a potent cell cycle arrest in a range of human epithelial cancer cell lines including colon, breast and lung cancer cells. Recently, Lord and colleagues identified a novel TNKS1/2 inhibitor (MSC2504877) and identified several synergistic targets through siRNA and drug-based screens, including CDK4/6[23]. We demonstrate that this drug synergy is absolutely dependent on suppression of canonical WNT signaling, and that the TNKS and CDK4 combined inhibition promotes enhanced cellular senescence. Our findings support recent work with an independent TNKS inhibitor and highlight the potential for WNT-targeted therapies to sensitize CRCs to FDA-approved CDK4/6 inhibitors.

## Results

### A kinome-focused screen identifies vulnerabilities in colorectal cancer

Tankyrase inhibition suppresses WNT signaling in diverse human and mouse cell types by stabilizing AXIN1 protein, resulting in decreased stability of beta-catenin and subsequently reduced transcription of TCF target genes[16, 17, 19–21, 24–26] (Fig 1A), yet TNKS inhibition alone is not sufficient to decrease cell proliferation in these cells under normal growth conditions[16, 20] (S1A Fig). However, in low serum conditions (2% FBS), cells become sensitive to TNKS inhibition[21] (Fig 1B). Similarly, suppression of MAP kinase activity via MEK inhibition synergizes with TNKS blockade[17] (Fig 1C). To identify other potential signaling dependencies that drive TNKS inhibitor resistance, we set out to evaluate the requirement for all human kinases, using a focused CRISPR-based loss-of-function screen.

We first identified a minimal dose of the TNKS inhibitor, XAV939[21] (hereafter XAV) that induced maximal AXIN1 stabilization (Fig 1D) and confirmed that treatment with this dose (1uM) maintained elevated AXIN1 levels up to 48hrs (S1B Fig). We then generated stable Cas9-expressing clones of the CRC line DLD1 (see Methods) and confirmed that each line (hereafter, LC9) could induce rapid depletion of sgRNAs targeting essential genes (e.g. *CDK1* or *RPA3*)[22]; We chose the three most efficient clones as replicates for the screen (S2A Fig).

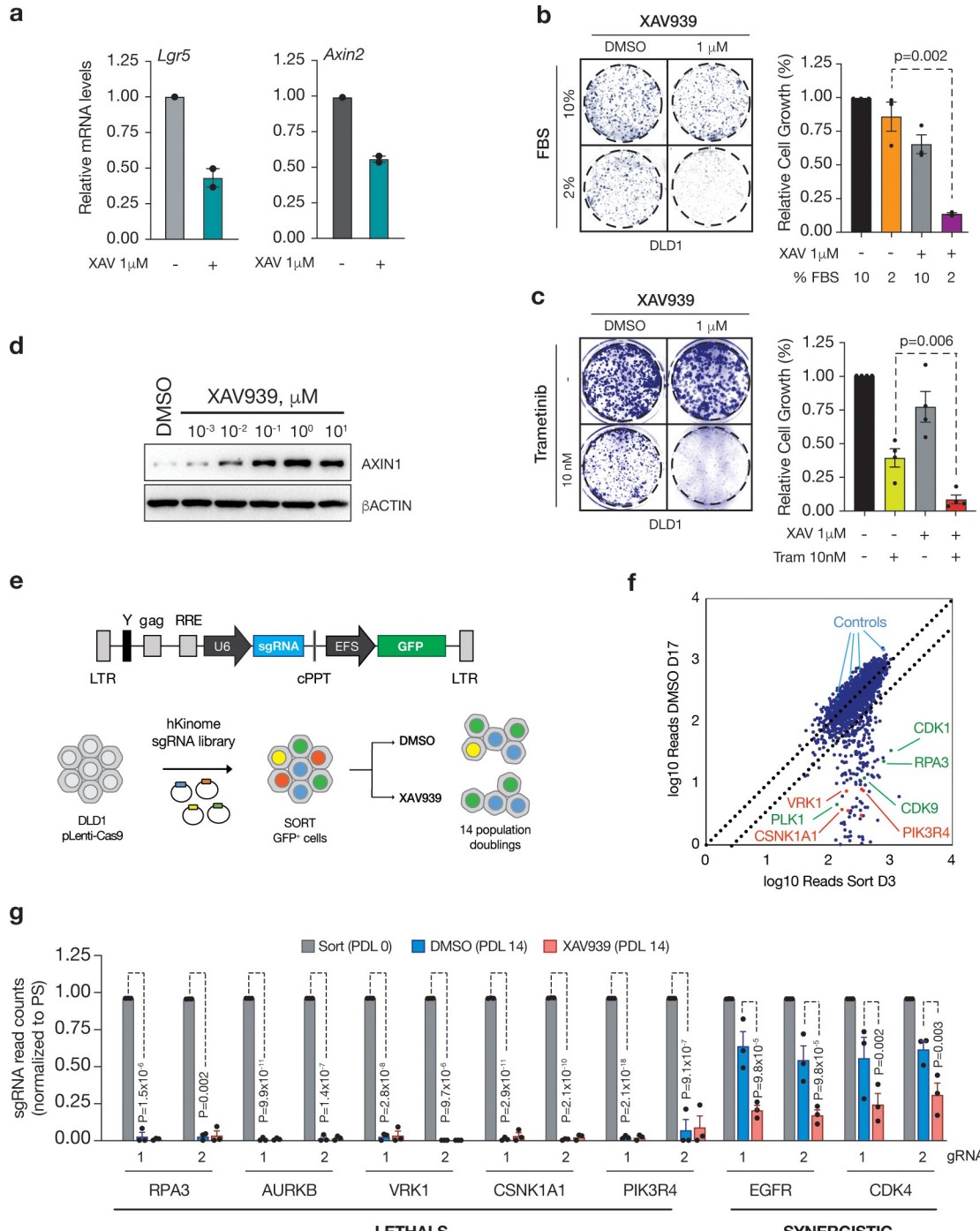

**Fig 1. A kinome-focused screen to identify novel vulnerabilities in colorectal cancer.** a. Relative mRNA levels measured by quantitative RT-PCR (qRT-PCR) in DLD1 cells treated with DMSO or XAV for 24h as indicated. b. Colony-forming assays in DLD1 cells grown in the indicated FBS and XAV concentrations (left); quantification of the experiment shown in (b) (n = 3 independent experiments, represented is mean and SEM). P values are shown (two-tailed Student's t test). c. Colony-forming assays in DLD1 cells grown in 10% FBS and the indicated Trametinib concentrations (left). Quantification of the experiment shown (right panel) (n = 4 independent experiments, represented is mean and SEM). P values are shown (two-tailed Student's t test). d. Western blot showing Axin1 stabilization upon DLD1 treatment (24h) with the indicated XAV concentrations. e. Schematic depiction of the backbone (LRG) in which the gRNA human kinome library was cloned (top). Outline of the human kinome gRNA library screen for XAV dependencies (bottom). f. Dot-plot representation of the log10 gRNA counts of DMSO-treated DLD1 cells at 14 population doublings (D17) vs sorted GFP-positive DLD1 cells at Day 3 post-gRNA library transduction (D3). Indicated are positive (green) and negative (blue) controls for gRNA depletion. Novel vulnerabilities are depicted in red

(CSNK1A1, VRK1, PIK3R4). g. Normalized gRNA counts at Sorting (gray), DMSO-PDL14 (blue) and XAV-PDL14 (red) in DLD1 cells. Results are shown normalized to sorting, n = 3 independent clones, as mean and SEM. Adjusted p values were calculated using DESeq. See S1 Table.

As expected, prolonged treatment of the clonal lines with 1uM XAV caused only a minor suppression in cell proliferation over three weeks in culture (containing 10% serum) (S2B Fig).

We next introduced a domain-focused kinome library targeting 482 human kinases (~6 sgRNAs/gene)[22] into each LC9 line with an average representation of 2000x (Fig 1E). Analysis of gRNA abundance in flow-sorted (GFP-positive) populations showed a high concordance between replicates with more than 99% of expected sgRNAs represented in each of the pools and good correlation across biological replicates (S1 Table; S2C Fig). Following expansion for 7 days to deplete sgRNAs targeting essential genes, each population was treated with either DMSO or 1uM XAV for 14 population doublings and relative abundance of individual sgRNAs was compared to post-sort (pre-treatment) samples (Fig 1E). As expected, control, neutral sgRNAs showed no significant change in abundance (Fig 1F and S2D and S3 Figs), while those targeting essential genes (e.g. CDK1, AURKB, CDK7) were strongly depleted in both DMSO and XAV conditions (S2 and S3 Tables, Figs 1F and 1G and 2). We also identified a subset of sgRNAs (VRK1, CSNK1A1, PIK3R4, SMG1, and BUB1) that were strongly depleted in DLD1 cells but not universally depleted in an array of cell lines assayed using an identical gRNA library[22] (Fig 2A). We confirmed some of these genes, including VRK1, CSNK1A1 and PIK3R4, were also essential in a range of other human CRC lines, suggesting that they may represent cell or tumor-type specific vulnerabilities (Fig 2B).

## Synthetic lethal interaction of Tankyrase inhibition with EGFR or CDK4/6 inhibition

We identified two genes that showed significant depletion (fold-change (FC) > 2, adjusted p-value < 0.05) of two or more sgRNAs in XAV-treated cells, compared to DMSO treatment (S3A Fig). These were epidermal growth factor receptor (EGFR), and cyclin-dependent kinase 4 (CDK4) (S4 Table and Fig 1G). Similar to MEK inhibitors, EGFR-targeting agents have been previously been linked to TNKSi resistance[19, 27] and we confirmed this synergy with multiple sgRNAs targeting EGFR, and the small molecule EGFR inhibitor, Gefitinib (Fig 3A–3C). We decided to focus on CDK4 because it had not been directly linked to TNKS or WNT pathway hyperactivation. Further, there are multiple clinically effective inhibitors targeting CDK4/6 (Palbociclib, Ribociclib, and Abemaciclib), though none are yet approved for CRC[28].

To assess how specific this synergy was for CDK4, we compared gRNA depletion across the entire CDK family in DMSO vs XAV-treated cells (Fig 4A). While some CDK genes are known essential fitness genes (i.e. CDK1, CDK7 or CDK9), other members had an intermediate impact (CDK2 and CDK4), and some were completely dispensable for DLD1 cell proliferation (CDK5, CDK6, and CDK10) (S3A Fig). Importantly, only CDK4 (and not the other closely related partner of Cyclin D, CDK6), showed enhanced depletion in the presence of XAV (S3A Fig). We validated the synergy between loss of CDK4 and TNKS inhibition using four independent CDK4 gRNAs targeting CDK4, all of which reduced cell fitness in fluorescence-based competitive assays (Fig 4B and 4C). Consistent with the results from the screen, depletion of CDK6 using two effective gRNAs only caused a minor disadvantage in competition assays, which was not significantly different between DMSO or XAV-treated cells (S3A Fig and Fig 4A–4C). These results highlight a specific CDK4 dependence upon TNKS inhibition in colorectal cancer cells.

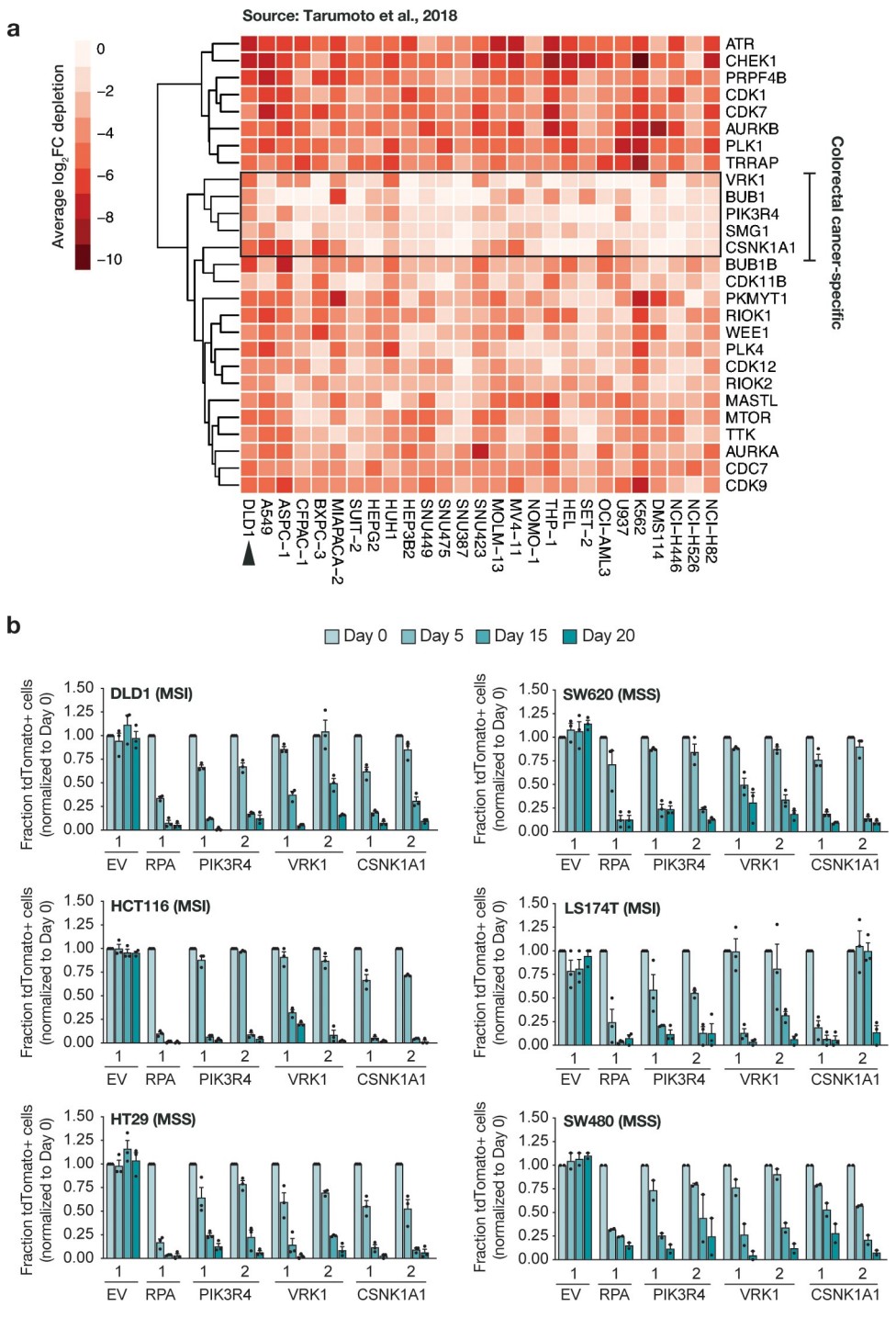

**Fig 2. New vulnerabilities in colorectal cancer.** a. Heatmap showing comparison of the gRNA depletion for potential lethal targets found in DLD1 (this study, arrowhead) vs the panel of cell lines described in Tarumoto et al, 2018. The colorectal-specific candidates are labelled. b. Validation of candidate colorectal-specific vulnerabilities in a panel of colorectal cancer cell lines from ATCC, through fluorescence-based competition assays. EV (empty gRNA control vector) and RPA are included as positive and negative controls, for reference of size effect (n = 3 independent clones for each cell line).

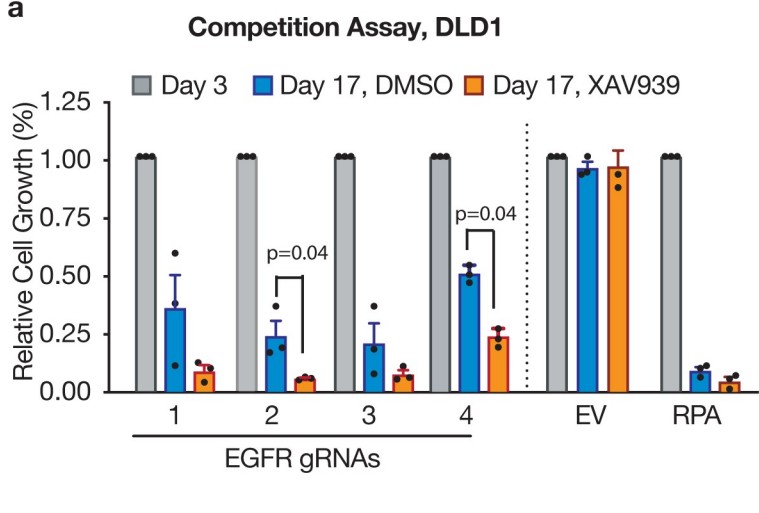

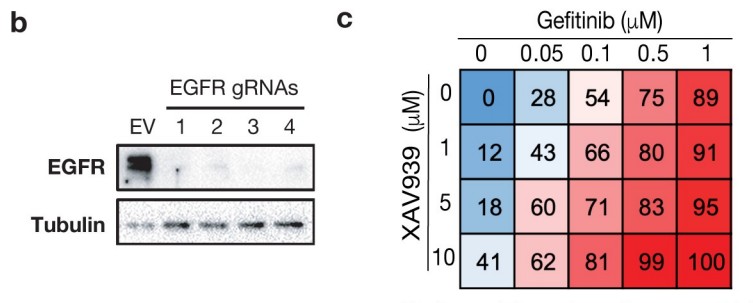

**Fig 3. Synergistic EGFR and TNKS inhibition.** a. Fluorescence-based competitive assay demonstrating enhanced inhibition of cell growth upon simultaneous TNKS inhibition and EGFR knock-out using four independent gRNAS. b. Western blot detection of EGFR in DLD1 cells treated with the indicated gRNAs. c. Cell growth inhibition in DLD1 cells treated with the indicated inhibitors for 15 days.

To confirm the synthetic lethality observed upon inhibition of TNKS together with gRNA-mediated CDK4 depletion was not specific to CRISPR-mediated CDK4 disruption, we treated DLD1 cells with varying concentrations of XAV and the CDK4/6 inhibitor Palbociclib. As expected, we observed increased suppression in colony formation following treatment with both drugs, that was greater than the additive effects of each treatment alone (Fig 4D and 4E). Conversely, to confirm the synergy observed was due to direct TNKS suppression and not inhibition of other PARP family members, we used two independent doxycycline(dox)-regulated shRNAs against TNKS1/2. As expected, shRNA-mediated depletion of TNKS sensitized DLD1 cells against Trametinib, Gefitinib and Palbociclib in the presence of dox (Fig 4F and 4G, S3B and S3C Fig). Finally, we further confirmed the on-target effect of TNKS inhibition in sensitizing cells to CDK4 blockade by treating cells with commercially-available small molecules that block TNKS activity via distinct mechanisms. TNKS inhibition through either the nicotinamide subsite (XAV939), adenosine subsite (G007-LK) or via a dual adenosine and nicotinamide binder (NVP-656), synergized with Palbociclib to suppress cell proliferation (Fig 4H).

The synergy between TNKS and CDK4 inhibition was not restricted to DLD1 cells, as we observed similar effects of TNKS silencing and Trametinib or Palbociclib treatment in the colorectal carcinoma cell line SW480 (bearing an APC truncating mutation, similar to DLD1)

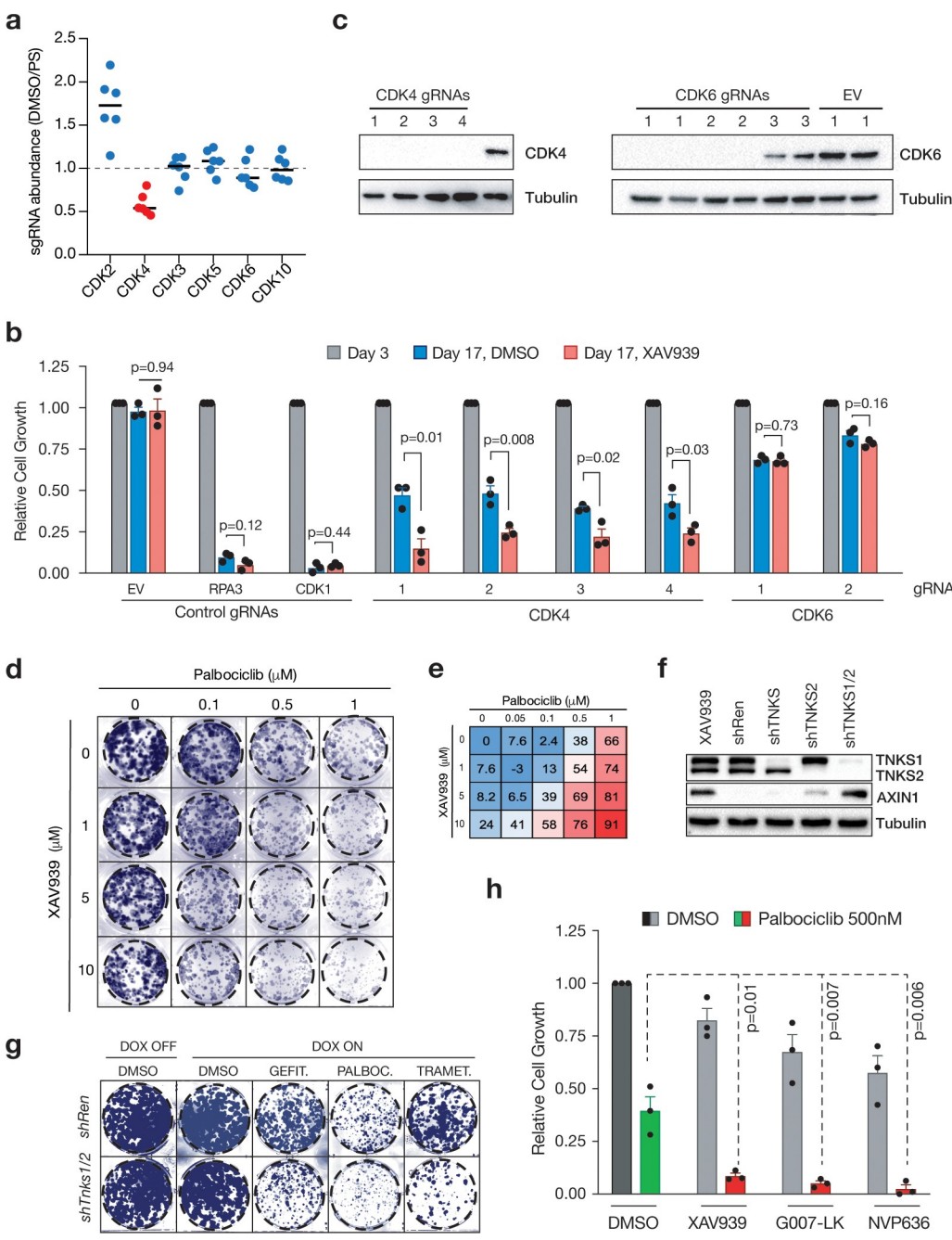

**Fig 4. Simultaneous inhibition of TNKS and CDK4 is synthetic lethal.** a. Normalized gRNA counts of Cyclin-dependent kinase (CDK) family members (DMSO D17 values, normalized to Sorting D3). b. Fluorescence-based gRNA competition assays by flow cytometry. Shown in mean and SEM, n = 3 independent DLD1 clones. P values were calculated using two-tailed Student's t test. EV = control empty gRNA vector. c. Western blot in DLD1-LC9 cells stably transduced with CDK4 or CDK6 gRNAS and selected in Puromycin and Blasticydin for 5 days. d. Colony forming assays of DLD1 cells grown in 10% FBS in the indicated inhibitor concentrations. e. Growth inhibition scores, shown as % of growth inhibition (normalized to non-treated conditions), of DLD1 cells in the indicated inhibitor concentrations. f. Western blot showing TNKS silencing with single or tandem shRNA constructs in DLD1 cells. Tandem TNKS silencing induces AXIN1 protein to a similar level as treatment with XAV939. g. Colony forming assays of DLD1 cells stably transduced with the indicated dox-inducible TNKS1/2 or Renilla shRNAs, respectively, and grown in colony-forming conditions in the presence of the indicated inhibitors +/- dox. TNKS1/2 shRNAs efficiency by western blot (bottom). g. Relative growth of DLD1 cells grown in colony-forming conditions in the presence of the indicated inhibitors +/- Palbociclib. N = 3 independent experiments, mean and SEM are shown (Student's t test).

(S3C Fig). Further, we saw enhanced growth suppression in multiple breast and lung cancer cell lines treated with XAV and Palbociclib, demonstrating a consistent combinatorial effect of TNKS and CDK4 blockade across a broad spectrum of cancer cell lines of epithelial origins (S3D Fig).

### Synergy of combined TNKS and CDK4/6 inhibition is dependent on WNT pathway suppression

TNKS has a number of PARsylation targets other than AXIN1/2, including telomere binding factor TRF1[29], the mitotic regulator NUMA1[30], and YAP/TAZ-associated angiomotin (AMOT) proteins[27], among others. To determine whether TNKS/CDK4 inhibitor synergy was dependent on suppression of WNT signaling we used CRISPR base editing to engineer an S45F missense mutation in endogenous CTNNB1[31] in DLD1 cells, rendering it insensitive to regulation by the APC destruction complex. We derived $CTNNB1^{S45F}$ DLD1 clones and treated them with XAV (1uM) and increasing doses of Palbociclib (10-1000nM). While XAV-treated parental cells were 3-fold more sensitive to Palbociclib (IC50 250 for XAV-treated cells and 800nM for DMSO-treated cells, respectively), $CTNNB1^{S45F}$ isogenic cells showed no change in response to treatment with XAV (Fig 5A–5D). Similarly, XAV-treatment of a natural $CTNNB1^{S45F}$ mutant cell line, HCT116, showed no increased sensitivity to Palbociclib, or genetic disruption of CDK4 by CRISPR (S4 Fig). These results suggest that TNKS-mediated sensitization to CDK4 inhibition is absolutely dependent on the ability of TNKS inhibitors to suppress WNT signaling. Importantly, we saw identical effects when XAV was combined with Trametinib or Gefitinib, implying that most reported drug synergies with TNKS inhibitors are likely mediated through WNT suppression (S4 Fig).

### Tankyrase inhibition enhances the cytostatic effects of Palbociclib

Palbociclib exerts its effects on target cells by blocking the activity of the D-type cyclin-dependent kinases CDK4/6 and subsequently inducing a G1 cell cycle arrest[32]. To understand how the combination of CDK4/6 and TNKS inhibition could lead to reduced cell growth in epithelial cells, we analyzed cell cycle kinetics in XAV and Palbociclib treated DLD1 cells. As expected, Palbociclib treatment caused a reduction in the proportion of cells in S-phase and a corresponding increase in the in the percentage of cells in G0/G1 (Fig 6A and 6B). Combined treatment with XAV caused a further drop in S-phase cells, resulting in more than 70% of cells arrested in G0/G1.

During analysis of the individual and combination treatments, we noted that XAV/Palbociclib treated cells showed a flattened morphology and increased intracellular vacuoles, characteristic of cellular senescence. Indeed, while XAV or Palbociclib only treated cells showed low senescence-associated beta-galactosidase (SA-ßGal) activity (5% and 17%, respectively; Fig 6C), combined XAV/Palbociclib treatment induced a 2.5-fold increase in SA-ßGal. Increased senescence was also observed in a panel of colorectal, breast and lung cancer cell lines upon combined therapy using XAV and Palbociclib (Fig 6D and 6E). Together, this data suggests that, combined inhibition of TNKS and CDK4 leads to enhanced cell cycle arrest and cellular senescence. It is not yet clear whether senescence-associated transcriptional changes and senescence-associated secretory phenotype (SASP) is consistently activated across different cell lines, or in vivo tumors.

## Discussion

Activation of the WNT pathway is a driver in many cancer types, particularly colorectal cancer. However, suppression of hyperactive WNT is not sufficient to induce cell cycle arrest in

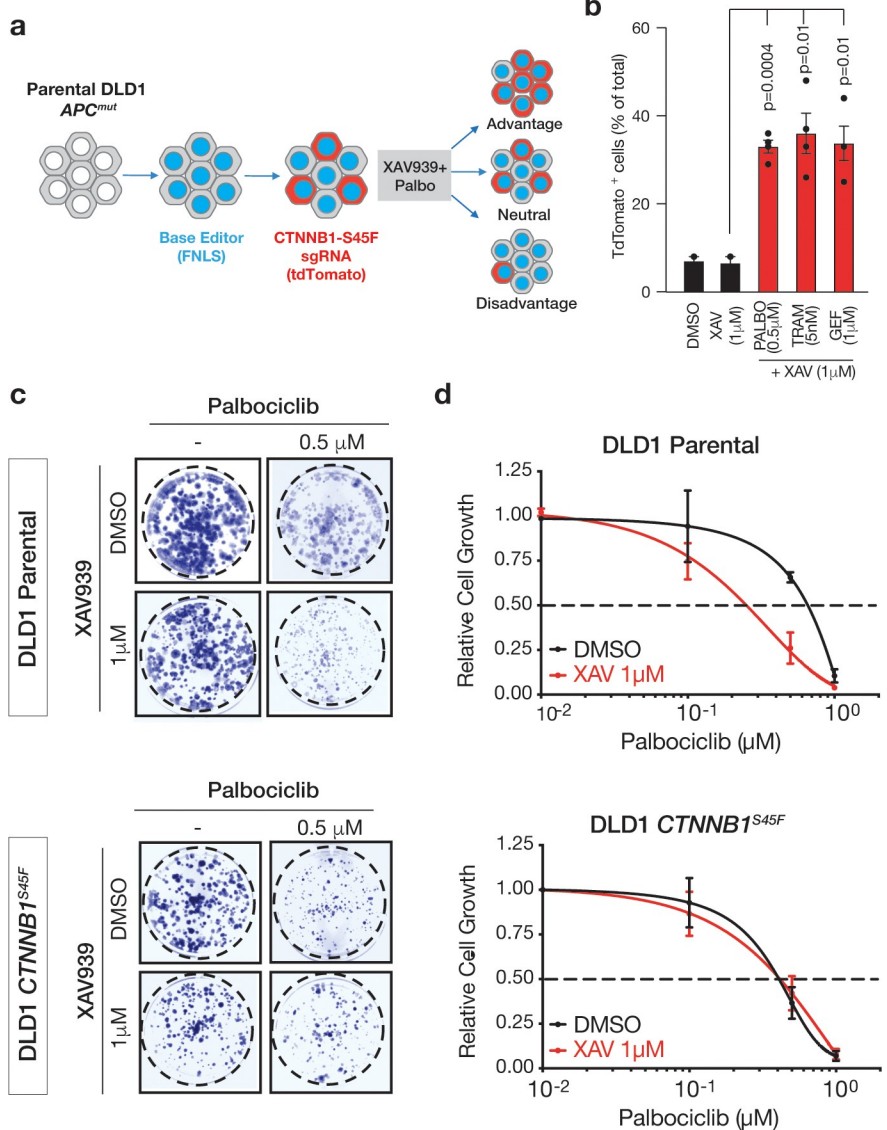

**Fig 5. CDK4 and TNKS synergy is dependent on canonical WNT signaling.** a. Schematic representation of competition assays, using a LRT2B backbone to track CTNNB1-S45F modified cells. b. Fluorescent competition assay showing the percentage of TdTomato-positive cells after 14 PDLs in the indicated drug concentrations. N = 4 independent experiments, p values were calculated using Student's t test. c. Colony-forming assays of parental (top panels) or S45F-edited (bottom panels) DLD1 cells, treated with the indicated drug concentrations. d. Relative cell growth, represented as % of DMSO-treated DLD1 cells, calculated from colony-forming assays cells of parental (top) or CTNNB1-S45F (bottom) backgrounds. Cells were treated with increasing Palbociclib concentrations in the absence (black lines) or presence of XAV. N = 3 independent assays, represented is mean and SD.

many human cancer cell lines. Using a domain-focused human kinome CRISPR screen, we sought to identify signaling dependencies that could enhance the sensitivity of CRC cells to WNT suppression by TNKS1/2 inhibition.

Previous *in vitro* and *in vivo* studies have shown that inhibition of either MEK1/2, EGFR, or AKT can enhance the effect of TNKS blockade[17, 19, 20]. To our surprise, we identified only two genes whose depletion synergized with TNKS inhibition: EGFR and CDK4. Though we did identify EGFR as a synthetic interaction with TNKS in DLD1 cells, neither MEK nor

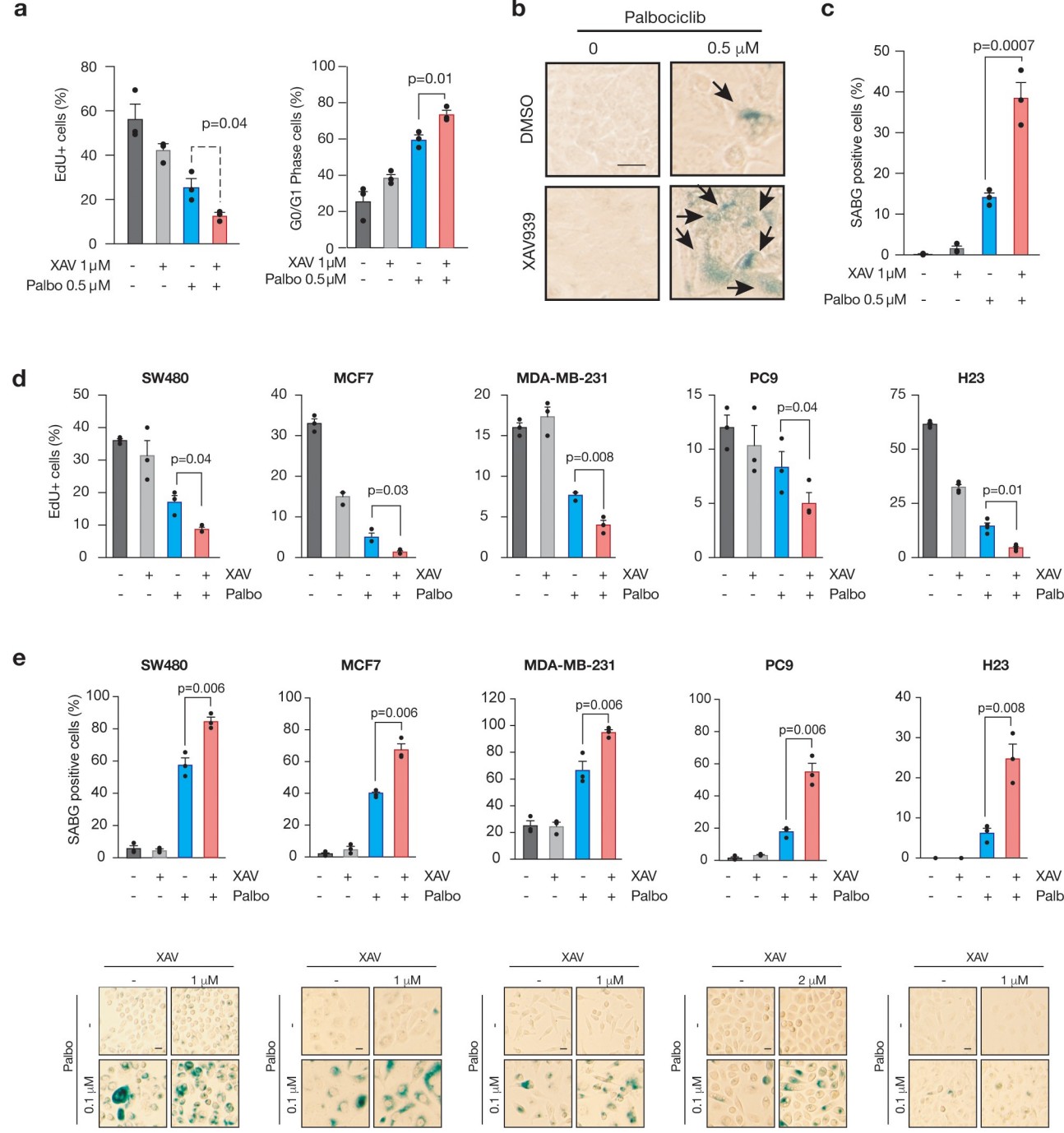

**Fig 6. TNKS inhibition reinforces CDK4 inhibition-mediated senescence.** a. Flow cytometry analysis of EdU incorporation (left) and EdU/PI G0/G1 arrest (right panel) in DLD1 cells treated for 15 days with the indicated inhibitors. Represented is mean and SEM, p values indicate two-way Student's t test (n = 3 independent experiments). b. Senescence-associated beta-galactosidase (SABG) staining in DLD1 cells, treated for 15 days, as indicated. Arrows indicate positive-scored cells. Scale bar, 25 microns. c. Quantification of the percentage of SABG-positive cells from the experiment shown in (b). N = 3 independent experiments, SEM and mean are depicted. P values correspond to two-sided Student's t test. d. Flow cytometry analysis of EdU incorporation in colon, breast and lung cell lines treated for 15 days with the indicated inhibitors. Represented is mean and SEM, p values indicate two-way Student's t test (n = 3 independent experiments). e. Quantification of the percentage of SABG-positive cells in colon, breast and lung cells. N = 3 independent experiments, SEM and mean are depicted. P values correspond to two-sided Student's t test (top panels). Representative images of the results shown in (d). Scale bars, 25 microns (bottom panels).

AKT scored. As DLD1 cells are sensitive to combined treatment with TNKS and MEK inhibitors (Fig 1C)[17], we expect that the failure to identify MEK or AKT is due to the functional redundancy of multiple MEK and AKT family members. This observation highlights the limitation of single sgRNA libraries for studying functionally redundant gene families, but also implies there may be other unidentified kinase targets that synergize with TNKS blockade.

Interestingly, our second hit (CDK4) is known to have a functionally redundant family member, CDK6. Yet, while CRISPR-mediated knockout of either CDK4 and CDK6 moderately reduced cell fitness, only CDK4 loss synergized with TNKS inhibition to block cell cycle progression, enhance apoptosis, and drive cell senescence. During the preparation of this manuscript, Menon and colleagues also identified an interaction between CDK4 and TNKS1/2 using a different small molecule TNKS inhibitor, MSC2504877[23]. Interestingly, in their siRNA-based screening platform, they also identified CDK6 as a synthetic vulnerability with TNKS inhibition. This apparent discrepancy in the role of CDK6 could be either due to the use of different CRC cell lines in our screens, or differences between siRNA and CRISPR-based approaches. Though, importantly, irrespective of reason underlying the difference, the data presented herein raise the possibility that potent CDK4-selective inhibitors would most likely enhance the response to TNKS/WNT inhibition, while potentially avoiding some of the on-target toxicity of dual CDK4/6 inhibitors. Though there are no selective CDK4 inhibitors yet commercially available, our data suggests they may prove effective in combination with WNT-targeting molecules.

In addition to identifying synergies with TNKS inhibition, our screen also revealed some possible CRC-selective dependencies. In particular, we identified a small set of genes (in particular VRK1, CSNK1A1, and PIK3R4) that were more dramatically depleted in colorectal cancer cell lines, than a range of leukemia, liver and pancreatic cancer cell lines[22]. There is limited *in vivo* data in the literature to assess whether these represent practical therapeutic targets, though mouse studies have shown that hypomorphic VRK1 animals are viable[33], and ablation of Csnk1a1 throughout the intestine does not disrupt normal homeostatic functions[34]. Further, while PIK3R4 deletion is embryonic lethal, it is not essential in all adult tissues[35]. Further work will be necessary to determine the requirement for these genes in normal and transformed adult tissues.

TNKS inhibitors are often considered direct WNT targeting drugs, though TNKS has many cellular targets in addition to AXIN1/2. In fact, Wang et al[27], recently reported that the synergy between TNKS and EGFR inhibition in lung cancer cells is due to reduction of YAP/TAZ-mediated signaling via regulation of AMOT proteins[27]. Our data in colorectal cancer cell lines, suggests that downstream induction of the WNT pathway by activating mutations in *CTNNB1* are sufficient to overcome the effects of TNKS blockade. Thus, at least in the CRC cells examined here, WNT pathway inhibition is a critical component of TNKS-mediated growth suppression. Further, while a number of the cell lines we assessed (e.g. MCF7, PC9) have no known WNT pathway activating mutations, they have been shown to respond to WNT-regulation [36–38]. However, given the breadth of TNKS targets, it is possible that genetic or cell type-specific oncogenic programs create differential dependencies on TNKS, or that the synergy observed between TNKS and CDK4 blockade is due to non-WNT dependent signaling. In all, the data imply that TNKS enzymes act to support oncogenic transformation through multiple independent mechanisms. Hence, while TNKS inhibitors may not always provide significant anti-cancer responses as single agents[17, 20], they may prove effective as part of specific treatment combinations, such as those described here and elsewhere[23].

Perhaps the most striking effect of combined TNKS/CDK4 inhibition was the robust induction of cell senescence. We observed this colon (DLD1 and SW480), breast (MCF7 and MDA231), and lung cancer (A549, H23 and PC9) cells, while Lord and colleagues observed a

near identical effect in an independent CRC cell line, COLO320DM, together suggesting it is not a cell-type specific phenomenon. The precise mechanism underlying the drug synergy that leads to enhanced senescence remains intriguing, although multiple published examples that WNT suppression blocks stemness programs and leads to profound chromatin changes that precede the induction of senescent phenotypes[39–41]. This notion aligns with our observation that downstream WNT activation can bypass the effect of combined TNKS/CDK4 inhibition. The enhanced induction of senescence, as opposed to cell cycle arrest or apoptosis, may have a beneficial impact in tumor therapy. Senescent cells secrete a range of inflammatory cytokines–the senescence associated secretory phenotype (SASP)–that can activate immune surveillance[42–45]. In some contexts, secreted factors from senescent cells recruit NK cells, resulting in tumor cell elimination[46–48], while in others, the infiltration of monocytes and granulocytes are the primary driver of senescent cell clearance[49]. In separate work, WNT hyperactivation has been directly linked to immunotherapy failure[50–52]. It is important to note that all of the work reported here has been performed in cancer cell lines *in vitro*, and it is not known whether the senescence phenotypes induced by combined TNKS/CDK4 inhibition will behave similarly *in vivo*. It will be important to determine whether combined suppression of WNT and CDK4 could provide potent tumor responses in vivo and as an effective combination with current immune checkpoint inhibitors.

In all, our results demonstrate some of the practical applications of focused CRISPR screens in identifying high-priority candidate liabilities. Follow up studies will demonstrate the practical applications in the clinic through co-clinical trials in matched patients and patient-derived organoids, in which a rapid pipeline could help identify vulnerabilities and to predict potential mechanisms of resistance to certain drug treatments.

## Methods

### Cell lines

Colon (DLD1, SW480, HCT116, HT29, LS174T, LoVo, T84 and SW620), breast (MDA-MB-231 and MCF7), and lung (A549, H23 and PC9) cell lines were purchased from ATCC, tested for mycoplasma and kept at low passage at all times (cells were discarded at passage >10). Colon and breast cancer cell lines were maintained in DMEM containing 10% Fetal Bovine Serum (FBS) and 1% Penicillin/Streptomycin (P/S). Lung cancer cell lines were grown in RPMI supplemented with 10% FBS and 1% P/S. All the experiments were performed at passage >10 after reception from the ATCC.

### Generation of Cas9-expressing clones

Cells were transduced with lentiviruses containing Cas9-P2a-Puro, selected for 72h in 2ug/ml Puromycin, and plated following serial dilutions in 96 well plates (5 plates per cell line). 24h later, wells containing a single cell were identified and picked when confluent, expanded and tested with control gRNAs (EV) or gRNAs targeting pan-essential genes such as *RPA3 or CDK1*[22]. Clones showing gRNA depletion kinetics similar to their parental counterparts through fluorescent competition assays were selected for further experimentation.

### CTNNB1 S45F base editing

We used puro-selected bulk FNLS-expressing DLD1 cells and further transduced them with a LRT2B-CTNNB1-S45F gRNA. After selection in 10ug/ml Blasticidin, we isolated clones and Sanger-sequenced the gDNA surrounding the mutation. S45F-confirmed clones were used for subsequent experiments.

## Pooled negative-selection human kinome crRNA-based screen

A domain-focused kinome gRNA library[22] was used for assessing novel vulnerabilities and TNKS-specific dependencies in CRC. This CRISPR pool was transduced into three DLD1 clones stably expressing SpCas9 using conditions that led predominantly to single lentiviral integration per cell and ensuring a representation of at least 6,000 cells per gRNA. 48h after transduction, cells were FACS-sorted for GFP expression. Half of the sorted cells (9 million transduced cells; 3,000 cells per gRNA) were snap-frozen and kept for sequencing as original library representation (Day 2), and the rest was allowed to recover for 7 days to ensure crRNA-based genome editing. At Day 9 post-transduction, sorted cells were plated either in the presence of DMSO or 1ug/ml XAV939, replacing the medium every 2 days and splitting the plates every 5 days while ensuring that at least 15 million cells were kept at each passage to maintain sufficient library representation. After 14 population doublings (17 days) in the presence of DMSO or XAV939, cells were snap-freeze collected as final timepoints (Day 17, PDL14). Genomic DNA was extracted from Day2, Day17-DMSO and Day17-XAV939 using QIAGEN DNA mini kit. 40 parallel PCRs were performed using 4ug of total gDNA as a template, pooled using PCR purification kit (QIAGEN) and eluted in 85ul PCR-grade water. After template repair, dA tailing, barcoding and pre-capture PCR as described[22], libraries were quality-checked using Bioanalyzer. 8pM of each barcoded library[22] was analyzed on Illumina MiSeq in pools of 4 samples per lane (MiSeq v2, 150x2 cycles). Two-by-two comparisons and adjusted p-values of gRNA reads were calculated using DESeq R package.

## Cumulative population doublings

5000 cells per well were seeded in 48-well plates and cultured for 20 days. Media was replaced every 2 days and cells were split, counted and seeded at 5000 cells per well every 5 days. Cumulative population doublings were calculated using the formula $PDL = X0 + 3.322(logY - LogI)$ were PDL stands for cumulative population doublings, X0 is the last number of doublings for a given timepoint, Y is the final number of cells and I is the number of cells plated.

## Drugs

XAV939, NVP636, G007-LK, Palbociclib, Gefitinib and Trametinib were purchased form Selleckchem. All drugs were aliquoted at 20mM stocks and stored at -80C indefinitely. 1mM aliquots were prepared from these stocks and stored at -20C for 6–12 months. 1mM aliquots were thawed once and aliquots in use were kept at 4C for a period <2 weeks (leftovers were discarded if left unused after these periods of time). The working concentrations are indicated in each Fig panel. Drugs in solution for cellular treatment were replenished every 48h by adding fresh media.

## Cloning

gRNAs were cloned into LRG or LRT2B lentiviral backbones using BsmBI restriction sites and adapters as described elsewhere[22, 31]. All the primers and plasmids are listed in S5 Table.

## Lentiviral transduction

293T cells were seeded at 90% confluence into 35mm dishes; 16h later, cells were transfected using PEI as follows: 5ug plasmid, 2.5ug PAX2 packaging vector and 1.25ug VSVg envelope vector were mixed into 150ul DMEM, vortexed 5 seconds and 30ul PEI were added and vortexed again; after 15min incubation at RT, transfection mix was added dropwise by swirling the 293T cell plates. 24h later, the medium was replaced with collection medium, and viral

supernatants were collected three times every 24h, pooled and preserved in 0.5ml aliquots at -80C indefinitely. Target cells were seeded at 50,000 cells per well of a 12-well plate, and 24h later the viral supernatant dilution containing 8ug/ml polybrene was added. 24h later the infection media was replaced with fresh media and cells were assayed or selected 48h post-transduction.

## Fluorescence-based competition assays

First, lentiviral supernatants were titrated for each cell line such that cells stably expressing Cas9 would achieve a 30–60% TdTomato+ or GFP+ cells at Day3 post-transduction. This timepoint was used for normalization of subsequent timepoints and these were represented as a fraction of Day3 infection efficiency (= 1). 1/20$^{th}$ of the cells was plated in flat-bottom 96 well plates and kept in culture by replacing the medium every 2 days. Cells were split every 5–7 days and fluorescence was assessed by flow cytometry using a BD Accuri C6 Plus or a Thermo-Fisher Attune NxT flow cytometer, both of them with 96 well plate adapters. To do so, cells were washed in 200ul of PBS, trypsinized in 50ul for 5min at 37C and resuspended in 200ul complete medium using a multichannel pippete. 1/20$^{th}$ of the cell suspension was transferred to a new plate and the rest was placed in a round-bottom 96 well plate compatible with the autosampler device of the flow cytometer.

## Senescence-associated beta-galactosidase staining

50,000 cells were seeded in 35mm dishes and pre-treated with each drug combo accordingly. 7 days later, cells were re-plated in 12 well plates at a density of 20.000 cells per well in the presence of drugs. 7 days later, cells were fixed and stained for beta-galactosidase activity using a kit from Cell Signaling Technologies (Cat. No. #9860). Increased b-galactosidase activity at pH6.0 detected by this approach reflects an increase in lysosomal β-D-galactosidase associated with oncogene-induced and replicative senescence [53]. Micrographs were quantified using Image J, and SABG-positive cells per field were normalized to the total number of cells per field, and at least four fields (>300 cells) per condition from three independent experiments were quantified. The drugs were replaced every 48h throughout the duration of the experiment.

## Cell cycle and EdU incorporation

EdU flow cytometry was performed using the Click-iT™ Plus EdU Alexa Fluor™ 647 Flow Cytometry Assay Kit (Thermo Fisher, #C10634). After 15 days of treatment, cells were subjected to EdU/PI staining following manufacturer's instructions.

## Western blot

200.000 cells were seeded in 35mm dishes and treated for the indicated times and conditions. Upon collection, cells were washed in 1ml cold PBS and scrapped off into 150ul RIPA buffer, incubated 15min on ice and centrifuged at 13.000rpm to collect protein lysates. 25ug per lane were loaded; antibodies used and dilutions were anti-Axin1 (CST, #2087), anti-Tnks1/2 (Santa Cruz, #sc-365897), anti-actin-HRP (Abcam, #ab49900), Anti-alpha-Tubulin (Millipore Sigma, #CP06), anti-CyclinD1 (Santa Cruz, #sc-450), anti-CDK4 (Santa Cruz, #sc-56277), anti-EGFR (Santa Cruz, #sc-373746) and anti-CDK6 (CST, #13331T).

## Colony-forming assays

Cells from a single-cell suspension were plated at a density of 1000 cells per well of a 12 well plate in 500ul of complete medium containing no drugs. Indicated treatments were started 24h after plating and refreshed every 48h unless indicated until the end of the experiment. After 15–25 days (when colonies were visible), cells were washed in PBS and fixed in a 4% PFA solution in PBS for 1h at room temperature, then rinsed in PBS and stained overnight in an orbital shaker using a 10% Giemsa solution in PBS. Then, plates were submerged in a distilled water solution, rinsed 3 times in water and dried overnight. Digital pictures were acquired with a BioRad Gel Doc device and pixel density was calculated using Adobe Photoshop. Experiments were performed 3 times independently, and values were normalized to DMSO controls.

## Supporting information

**S1 Fig. On target TNKS inhibition is not effective as monotherapy.** a. Colony forming assay quantification (top panel) and representative images (bottom panel) of cells grown in the indicated inhibitor concentrations (n = 2). b. Western blot detection of Axin1 accumulation at different incubation timepoints using the indicated XAV doses.
(TIF)

**S2 Fig. Domain-focused human kinome screen in colorectal cancer cells.** a. Fluorescence competition assays of DLD1 clones stably expressing pLenti-Cas9 and further transduced with RPA3 gRNA or a control gRNA empty vector. b. Cell growth by cumulative population doubling estimation in DLD1 cells treated as indicated with 1mM XAV or DMSO (n = 3). c. Dot plots of the gRNA reads in two-by-two comparisons. Correlation coefficients (R2) across conditions are shown. d. Volcano plot of Log10 adjusted p-values vs log2 fold-change in gRNA representation in DMSO D17 vs post-sorting cells (D3, PS). In red, gRNAs significantly depleted in DMSO. e. Normalized control (non-targeting) gRNA counts in DMSO (D17) vs XAV (D17)-treated cells.
(TIF)

**S3 Fig. Synergistic CDK4/6 and TNKS inhibition in multiple epithelial cell types.** a. Left panel, dot plot showing log10 Adj p values and log2 fold change of DMSO (D17) vs XAV (D17) in DLD1 cells. Significantly represented gRNAs are highlighted in red. Right panel, relative gRNA abundance of human CDK protein members at D17 (DMSO) vs post-sorting (PS) samples. b. Quantification of the colony forming assay shown in Fig 2F. DLD1 cells stably expressing inducible shRNAs against TNKS were treated with the indicated drugs +/- dox. c. Fluorescent competition assays in SW480 cells stably expressing shRNAs against TNKS1/2, treated with Trametinib (left) or Palbociclib (right) +/- dox. The GFP positive cells represent the proportion of shRNA-expressing fraction of each population, relative to D2 post-transduction. d. Colony forming assays (bottom panels) and quantification (top panels) of XAV and Palbociclib combinations as indicated, for a panel of epithelial cells lines, including lung and breast. N = 2–3 independent experiments, and p values represent Student's t test.
(TIF)

**S4 Fig. Canonical WNT signaling determines XAV-sensitization.** a. Fluorescent competition assay in HCT116 clones stably expressing Cas9 and further transduced with the indicated gRNAs, in the presence or absence of XAV. N = 3 clones, Student's t test b. HCT116 cells treated with the indicated doses of XAV and Palbociclib were seeded in colony-forming assays. c. Quantification of cell proliferation inhibition in HCT116 cells treated with the indicated

concentrations of Palbociclib or Gefitinib +/-XAV. d. shRNA-mediated knock-down of TNKS does not influence cellular sensitivity to Palbociclib, Trametinib or Gefitinib in HCT116 cells. e. Quantification of the experiment shown in (d). f. Trametinib sensitivity in DLD1 parental or base-editing-generated S45F mutant isogenic DLD1 cell lines.
(TIF)

**S1 Table. List of raw gRNA counts.**
(CSV)

**S2 Table. DESeq analysis of DMSO vs Post-sorting (PS) samples.**
(CSV)

**S3 Table. DESeq analysis of XAV vs Post-sorting (PS) samples.**
(CSV)

**S4 Table. DESeq analysis of DMSO vs XAV samples.**
(CSV)

**S5 Table. List of primers used in this study.**
(CSV)

## Acknowledgments

We would like to thank Weill Cornell Flow Cytometry Core and Weill Cornell Genomics Facility for their experimental support; Edward Kastenhuber, Elena Piskounova, Dawid Nowak, Lucia Morgado, and Dow Lab members for critical reading of the manuscript and discussions. The content is solely the responsibility of the authors and does not necessarily represent the official views of the NIH.

## Author Contributions

**Conceptualization:** Miguel Foronda, Christopher R. Vakoc, Lukas E. Dow.

**Data curation:** Miguel Foronda, Yusuke Tarumoto, Jill Zimmerman, Lukas E. Dow.

**Formal analysis:** Miguel Foronda, Yusuke Tarumoto, Lukas E. Dow.

**Funding acquisition:** Lukas E. Dow.

**Investigation:** Miguel Foronda, Emma M. Schatoff, Benjamin I. Leach, Bianca J. Diaz, Jill Zimmerman, Sukanya Goswami, Michael Shusterman, Lukas E. Dow.

**Methodology:** Miguel Foronda, Yusuke Tarumoto, Jill Zimmerman, Christopher R. Vakoc, Lukas E. Dow.

**Project administration:** Lukas E. Dow.

**Resources:** Christopher R. Vakoc, Lukas E. Dow.

**Supervision:** Miguel Foronda, Lukas E. Dow.

**Validation:** Miguel Foronda.

**Visualization:** Miguel Foronda.

**Writing – original draft:** Miguel Foronda, Lukas E. Dow.

**Writing – review & editing:** Miguel Foronda, Yusuke Tarumoto, Lukas E. Dow.

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
