## [Decision Letter · Decision Letter 0]

6 Aug 2019

PONE-D-19-18136

Tankyrase inhibition sensitizes cells to CDK4 blockade

PLOS ONE

Dear Dr Dow,

Thank you for submitting your manuscript to PLOS ONE. After careful consideration, we feel that it has merit but does not fully meet PLOS ONE’s publication criteria as it currently stands. Therefore, we invite you to submit a revised version of the manuscript that addresses the points raised during the review process.

Two experts have reviewed the manuscript and found that this current study was well written and compelling, but the reviewers commented that the study can be more in details as requested and in vivo aspects can be more discussed. Further, the authors can move the supplementary figure to the main figures for easier access by readers.

We would appreciate receiving your revised manuscript by Sep 20 2019 11:59PM. To enhance the reproducibility of your results, we recommend that if applicable you deposit your laboratory protocols in protocols.io, where a protocol can be assigned its own identifier (DOI) such that it can be cited independently in the future. For instructions see: http://journals.plos.org/plosone/s/submission-guidelines#loc-laboratory-protocols

We look forward to receiving your revised manuscript.

Kind regards,

Jung Weon Lee, Ph.D.

Academic Editor

PLOS ONE

Journal Requirements:

LED is a scientific advisory board member and stockholder in Mirimus Inc., who have licensed shRNA technology related to this study. All other authors declare no competing financial interests.

3. Our staff editors have determined that your manuscript is likely within the scope of our Targeted Anticancer Therapies and Precision Medicine Call for Papers. This editorial initiative is headed by a team of Guest Editors for PLOS ONE: Andrew Cherniack, Anette Duensing, Steven Gray, Sunil Krishnan, Chandan Kumar-Sinha and Gayle Woloschak. The Collection will encompass a diverse range of research articles about the identification and classification of driver genes and somatic alterations, target and drug discovery, mechanisms of drug resistance, and early detection and screening.  Additional information can be found on our announcement page: https://collections.plos.org/s/targeted-anticancer-therapies.

If you would like your manuscript to be considered for this collection, please let us know in your cover letter and we will ensure that your paper is treated as if you were responding to this call. If you would prefer to remove your manuscript from collection consideration, please specify this in the cover letter.

4. Please include your tables as part of your main manuscript and remove the individual files. Please note that supplementary tables (should remain/ be uploaded) as separate "supporting information" files

Reviewers' comments:

Reviewer's Responses to Questions

**Comments to the Author**

1. Is the manuscript technically sound, and do the data support the conclusions?

Reviewer #1: Yes

Reviewer #2: Yes

2. Has the statistical analysis been performed appropriately and rigorously? 

Reviewer #1: Yes

Reviewer #2: Yes

3. Have the authors made all data underlying the findings in their manuscript fully available?

Reviewer #1: Yes

Reviewer #2: Yes

4. Is the manuscript presented in an intelligible fashion and written in standard English?

Reviewer #1: Yes

Reviewer #2: Yes

5. Review Comments to the Author

Reviewer #1: Tankyrase 1/2 positively regulate Wnt signaling by targeting the beta catenin destruction complex. However, TNKS inhibition show limited anti-proliferative capacity in human cancer cell lines. In this manuscript, the authors used a kinome-focused CRISPR screen to identify potential drugs that synergizes with TNKS inhibition. They found that loss of CDK4 synergizes with TNKS inhibition leading to cell cycle arrest at the G1 phase and the cells eventually senesced. They then showed that this synergy is dependent on suppression of canonical Wnt signaling through CRISPR base editing to introduce an S45F mutation in CTNNB1 in DLD1 which renders CTNNB1 insensitive to regulation to the APC destruction complex. Finally, they provided evidence that combined TNKS and CDK4 inhibition was effective in driving cellular senescence in a variety of epithelial tumor cell types.

Overall the authors showed quite compelling data that synergistic interaction between TNKS and CDK4 inhibition is effective across many cancer types.

A major limitation of the study is that it is entirely conducted in culture. The authors should note this. Whether this synergy exists in more predictive models such as a patient derived xenograft or an orthotopic tumor is not established here. For example, Wnt pathway inhibition has recently been shown to have impressive effects on cell cycle progression in vivo but not in vitro. Therefore in vivo the additive effect of Wnt and CKD4 inhibition might not be present. This should be discussed.

The efficacy in multiple cell lines (Fig 4) that are not known to be mutant in the Wnt/β‑catenin pathway suggests this is a general phenomenon. It would be useful to know (or speculate in the discussion) if the senescence phenotype in those cells (e.g. MCF7, PC9) also requires degradation of β‑catenin, or is due to another effect of tankyrase inhibition?

I am not aware there is a page limit for PLOS ONE; it would be useful to put the data in the manuscript and move it out of supplementary data. There are four supplementary figures cited before we get to figure 2.

Reviewer #2: This is a well written and presented manuscript that clearly demonstrates a synthetic interaction between tankyrase inhibition and loss or reduced function of Cdk4.

Apart from two typographical errors in the figure legends of Supplementary Figures 3 and 4 I only have minor suggestions to improve the manuscript.

1. Please describe the nature of the senescence associated b-galactosidase activity.

2. Were any assays conducted for molecules normally secreted by senescent cells?

3. The authors state that neither MEK nor AKT were detected in the CRISPR screen but pharmacological inhibition of both has been shown to enhance TNKS blockers. It is possible that gRNAs for these were ineffectual. Were MEK or AKT tested individually for interaction with TNKS via separate gRNAs or siRNAs?

6. PLOS authors have the option to publish the peer review history of their article (what does this mean?). If published, this will include your full peer review and any attached files.

Reviewer #1: No

Reviewer #2: No

---

## [Author Response · Author response to Decision Letter 0]

18 Nov 2019

Response to Reviewers' comments:

We thank the reviewers for their positive and thoughtful responses. Below we outline changes made to the manuscript to address the reviewers’ concerns. In short, we have restructured the manuscript to present more of the data in main figures and added addition discussion to more clearly highlight the limitations of the some of the experiments described. We hope the revised manuscript is suitable for publication in PLoS One.

Reviewer #1: 

Tankyrase 1/2 positively regulate Wnt signaling by targeting the beta catenin destruction complex. However, TNKS inhibition show limited anti-proliferative capacity in human cancer cell lines. In this manuscript, the authors used a kinome-focused CRISPR screen to identify potential drugs that synergizes with TNKS inhibition. They found that loss of CDK4 synergizes with TNKS inhibition leading to cell cycle arrest at the G1 phase and the cells eventually senesced. They then showed that this synergy is dependent on suppression of canonical Wnt signaling through CRISPR base editing to introduce an S45F mutation in CTNNB1 in DLD1 which renders CTNNB1 insensitive to regulation to the APC destruction complex. Finally, they provided evidence that combined TNKS and CDK4 inhibition was effective in driving cellular senescence in a variety of epithelial tumor cell types.

Overall the authors showed quite compelling data that synergistic interaction between TNKS and CDK4 inhibition is effective across many cancer types.

A major limitation of the study is that it is entirely conducted in culture. The authors should note this. Whether this synergy exists in more predictive models such as a patient derived xenograft or an orthotopic tumor is not established here. For example, Wnt pathway inhibition has recently been shown to have impressive effects on cell cycle progression in vivo but not in vitro. Therefore in vivo the additive effect of Wnt and CKD4 inhibition might not be present. This should be discussed.

The reviewer is, of course, correct that it is difficult to extrapolate these in vitro findings to in vivo response. We have included additional discussion in the second last paragraph that mentions potential implications to tumor treatment. As follows:

“It is important to note that all of the work reported here has been performed in cancer cell lines in vitro, and it is not known whether the senescence phenotypes induced by combined TNKS/CDK4 inhibition will behave similarly in vivo. It will be important to determine whether combined suppression of WNT and CDK4 could provide potent tumor responses in vivo and as an effective combination with current immune checkpoint inhibitors.” 

The efficacy in multiple cell lines (Fig 4) that are not known to be mutant in the Wnt/β‑catenin pathway suggests this is a general phenomenon. It would be useful to know (or speculate in the discussion) if the senescence phenotype in those cells (e.g. MCF7, PC9) also requires degradation of β‑catenin, or is due to another effect of tankyrase inhibition?

It is indeed possible that the synergy observed between TNKS inhibition and CDK4 inhibition in some of the cell lines is not WNT-dependent. Unfortunately, we were unable to identify reliable clones activating CTNNB1 mutations in other cell lines we assessed, to directly address this point. Instead, we have expanded our discussion to highlight the limitations of our current analysis, and possible other explanations. New text in the discussion is as follows:

“TNKS inhibitors are often considered direct WNT targeting drugs, though TNKS has many cellular targets in addition to AXIN1/2. In fact, Wang et al27, recently reported that the synergy between TNKS and EGFR inhibition in lung cancer cells is due to reduction of YAP/TAZ-mediated signaling via regulation of AMOT proteins27. Our data in colorectal cancer cell lines, suggests that downstream induction of the WNT pathway by activating mutations in CTNNB1 are sufficient to overcome the effects of TNKS blockade. Thus, at least in the CRC cells examined here, WNT pathway inhibition is a critical component of TNKS-mediated growth suppression. Further, while a number of the cell lines we assessed (e.g. MCF7, PC9) have no known WNT pathway activating mutations, they have been shown to respond to WNT-regulation {Zhang, 2015 #3119;Lamb, 2013 #3120;Schlange, 2007 #3121}. However, given the breadth of TNKS targets, it is possible that genetic or cell type-specific oncogenic programs create differential dependencies on TNKS, or that the synergy observed between TNKS and CDK4 blockade is due to non-WNT dependent signaling. In all, the data imply that TNKS enzymes act to support oncogenic transformation through multiple independent mechanisms.”

I am not aware there is a page limit for PLOS ONE; it would be useful to put the data in the manuscript and move it out of supplementary data. There are four supplementary figures cited before we get to figure 2.

As the reviewer suggests, we have reorganized the manuscript to make more of the data available in the main figures.

Reviewer #2: 

This is a well written and presented manuscript that clearly demonstrates a synthetic interaction between tankyrase inhibition and loss or reduced function of Cdk4.

Apart from two typographical errors in the figure legends of Supplementary Figures 3 and 4 I only have minor suggestions to improve the manuscript.

1. Please describe the nature of the senescence associated b-galactosidase activity.

This is a standard assay for identifying senescent cells. It measures lysosomal activity of β‐D‐galactosidase at a specific pH (pH6.0). We used a commercial kit to measure this, and have included a more descriptive overview of the assay in the methods, as follows:

“Senescence-Associated Beta-Galactosidase Staining

50,000 cells were seeded in 35mm dishes and pre-treated with each drug combo accordingly. 7 days later, cells were re-plated in 12 well plates at a density of 20.000 cells per well in the presence of drugs. 7 days later, cells were fixed and stained for beta-galactosidase activity using a kit from Cell Signaling Technologies (Cat. No. #9860). Increased b-galactosidase activity at pH6.0 detected by this approach reflects an increase in lysosomal β‐D‐galactosidase associated with oncogene-induced and replicative senescence {Lee, 2006 #3123}.”

2. Were any assays conducted for molecules normally secreted by senescent cells?

We performed some transcript analysis on 2 cell lines (DLD1 and PC9), measuring the abundance of previously reported markers of senescence (IL6, IL8 and CDKN1A). In general, the induction of mRNA transcripts was far more variable than the SA�-gal assay (example data shown below). 

While we see some of the combination treatment samples have high or higher levels of these transcripts, there is a lot of sample-to-sample (and collection-to-collection) variability. In contrast to the rest of the data in the manuscript, we feel this data is not robust, likely owing to a number of technical variables we have not resolved. Given this, we believe this data does not contribute significantly to the overall manuscript, and may confuse rather than inform. We do agree that determining whether TNKSi/CDK4i-treated cells induce the SASP is an important question, just not one that we feel comfortable claiming anything about – in either direction – with current data/experiments. In place of convincing experimental data, we have included a statement at the end of the results as follows:

“Together, this data suggests that, combined inhibition of TNKS and CDK4 leads to enhanced cell cycle arrest and cellular senescence. It is not yet clear whether senescence-associated transcriptional changes and senescence-associated secretory phenotype (SASP) is consistently activated across different cell lines, or in vivo tumors.”

We are happy to amend this if the reviewer believes it does not convey the appropriate message.

3. The authors state that neither MEK nor AKT were detected in the CRISPR screen but pharmacological inhibition of both has been shown to enhance TNKS blockers. It is possible that gRNAs for these were ineffectual. Were MEK or AKT tested individually for interaction with TNKS via separate gRNAs or siRNAs?

We did not test individual MEK or AKT sgRNAs. There was one sgRNA for both MEK1 and MEK2 that was strongly depleted (with or without drug), but we cannot rule out the possibility that this was due to an off-target effect (https://www.biorxiv.org/content/10.1101/809970v1). Still a single sgRNA would not have met our screen criteria to be called a hit. 

We believe the reason for missing these validated targets in the screen is most functional redundancy. Depletion of either MEK1 or MEK2 alone (or any of the 3 AKT family members) would likely not be sufficient to disable signaling through these pathways. Indeed, we were surprised to identify CDK4, given it is largely redundant with CDK6, though that selectively was one of the exciting findings of this work. The passage in the second paragraph of the discussion that highlights this issue is as follows:

“Though we did identify EGFR as a synthetic interaction with TNKS in DLD1 cells, neither MEK nor AKT scored. As DLD1 cells are sensitive to combined treatment with TNKS and MEK inhibitors (Fig 1c)17, we expect that the failure to identify MEK or AKT is due to the functional redundancy of multiple MEK and AKT family members. This observation highlights the limitation of single sgRNA libraries for studying functionally redundant gene families, but also implies there may be other unidentified kinase targets that synergize with TNKS blockade.”

---

## [Decision Letter · Decision Letter 1]

4 Dec 2019

Tankyrase inhibition sensitizes cells to CDK4 blockade

PONE-D-19-18136R1

Dear Dr. Dow,

We are pleased to inform you that your manuscript has been judged scientifically suitable for publication and will be formally accepted for publication once it complies with all outstanding technical requirements.

With kind regards,

Jung Weon Lee, Ph.D.

Academic Editor

PLOS ONE

Additional Editor Comments (optional):

Reviewers' comments:

Reviewer's Responses to Questions

**Comments to the Author**

1. If the authors have adequately addressed your comments raised in a previous round of review and you feel that this manuscript is now acceptable for publication, you may indicate that here to bypass the “Comments to the Author” section, enter your conflict of interest statement in the “Confidential to Editor” section, and submit your "Accept" recommendation.

Reviewer #1: All comments have been addressed

2. Is the manuscript technically sound, and do the data support the conclusions?

Reviewer #1: Yes

3. Has the statistical analysis been performed appropriately and rigorously? 

Reviewer #1: Yes

4. Have the authors made all data underlying the findings in their manuscript fully available?

Reviewer #1: Yes

5. Is the manuscript presented in an intelligible fashion and written in standard English?

Reviewer #1: Yes

6. Review Comments to the Author

Reviewer #1: (No Response)

7. PLOS authors have the option to publish the peer review history of their article (what does this mean?). If published, this will include your full peer review and any attached files.

Reviewer #1: No

---

## [Editor Report · Acceptance letter]

20 Dec 2019

PONE-D-19-18136R1 

Tankyrase inhibition sensitizes cells to CDK4 blockade 

Dear Dr. Dow:

I am pleased to inform you that your manuscript has been deemed suitable for publication in PLOS ONE. Congratulations! Your manuscript is now with our production department. 

With kind regards,

on behalf of

Dr. Jung Weon Lee 

Academic Editor

PLOS ONE